# Entropy-Driven Pre-Tokenization for Byte-Pair Encoding

**Yifan Hu** [* 1]  **Frank Liang** [* 1]  **Dachuan Zhao** [* 1]  **Jonathan Geuter** [1 2]  **Varshini Reddy** [3]  **Craig W. Schmidt** [3]  **Chris Tanner** [3]

## Abstract

Byte-Pair Encoding (BPE) has become a widely adopted subword tokenization method in modern language models due to its simplicity and strong empirical performance across downstream tasks. However, applying BPE to unsegmented languages such as Chinese presents significant challenges, as its frequency-driven merge operation is agnostic to linguistic boundaries. To address this, we propose two entropy-informed pre-tokenization strategies that guide BPE segmentation using unsupervised information-theoretic cues. The first approach uses pointwise mutual information and left/right entropy to identify coherent character spans, while the second leverages predictive entropy derived from a pretrained GPT-2 model to detect boundary uncertainty. We evaluate both methods on a subset of the PKU dataset and demonstrate substantial improvements in segmentation precision, recall, and F1 score compared to standard BPE. Our results suggest that entropy-guided pre-tokenization not only enhances alignment with gold-standard linguistic units but also offers a promising direction for improving tokenization quality in low-resource and multilingual settings.

## 1. Introduction

Modern large language models (LLMs) often rely on BPE as a core tokenization strategy because it is simple and effective, leading to its widespread adoption (Sennrich et al., 2015). BPE iteratively merges frequent character pairs to construct a compact vocabulary, which enables the model to capture meaningful subword units and represent a wide range of linguistic phenomena across different languages. Its success in English and many Indo-European languages has made it the default tokenizer in many large-scale models such as GPT (Radford et al., 2019a), BERT (Devlin et al., 2019), and RoBERTa (Liu et al., 2019).

However, the application of BPE to Chinese presents unique challenges. Unlike alphabetic languages, Chinese lacks explicit word boundaries (e.g., spaces), and each character can serve as a standalone word or part of multi-character words with varying syntactic or semantic roles (Sproat et al., 1994). When BPE is applied naively, treating each character as a base unit and relying solely on frequency-driven merging, it often fails to capture the true internal structure of Chinese words. As a result, the created token sequences may not align with linguistically meaningful units, which can degrade downstream performance and interpretability. To this end, we introduce and evaluate two distinct entropy-driven pre-tokenization strategies for BPE (cmp. Section 3):

- **Statistical Methods:** We compute pointwise mutual information (PMI) (Church & Hanks, 1990) and left/right entropy to identify potential segmentation boundaries based on local co-occurrence strength and contextual diversity.

- **Auto-regressive LLM-based Methods:** We use a pretrained GPT-2 model (Radford et al., 2019a) to estimate token-level predictive entropy, leveraging model uncertainty to guide boundary detection.

We examine each approach independently and analyze their effect on BPE vocabulary learning and downstream segmentation quality. We compare both entropy-informed BPE variants to a standard frequency-driven BPE baseline, highlighting differences in tokenization granularity, compression efficiency, and alignment with gold-standard Chinese word segmentation. Our findings demonstrate that incorporating entropy in pre-tokenization can reshape BPE token structure and outperform naive BPE with respect to human-annotated gold-standard segmentation boundaries, offering new insights into subword modeling in unsegmented scripts for languages without native whitespace-denoted word boundaries like Chinese.

*Equal contribution [1]Harvard University, Cambridge, MA [2]Kempner Institute, Allston, MA [3]Kensho Technologies, Cambridge, MA. Correspondence to: Yifan Hu <yifan_hu@fas.harvard.edu>, Frank Liang <fliang@fas.harvard.edu>, Dachuan Zhao <dachuan_zhao@fas.harvard.edu>.

*Proceedings of the ICML 2025 Tokenization Workshop (TokShop)*, Vancouver, Canada. PMLR 267, 2025. Copyright 2025 by the author(s).

## 2. Related Works

### 2.1. Subword Tokenization via BPE

Byte-Pair Encoding (BPE) was introduced in data-compression research (Gage, 1994), then repurposed for open-vocabulary neural translation (Sennrich et al., 2015). As a greedy algorithm, BPE begins with an initial vocabulary of individual characters. At each iteration, it identifies the most frequent pair of symbols $(x, y)$ that occur adjacently in the corpus and merges them into a new symbol $z = xy$. The corpus is then updated by replacing all the occurrences of $(x, y)$ with $z$, and the process repeats until a target vocabulary size is reached. The core pair-finding and merging operation of BPE is computationally efficient, since it only considers bigrams at each step. As for vocabulary robustness, initialization with individual characters guarantees complete vocabulary coverage, thereby eliminating out-of-vocabulary issues. Subsequently, the iterative merge process captures frequent subword patterns, effectively enhancing the representation with more informative subword units. Consequently, BPE is widely used as a tokenization method in many large language models (Radford et al., 2019a; Liu et al., 2019; Touvron et al., 2023).

### 2.2. Pre-tokenization Constraints

Most subword algorithms, including BPE, operate on pretokenized input sequences, where initial boundaries, typically defined by whitespace, act as hard constraints on the merge process. In fact, only very recently has research investigated removing these constraints (Schmidt et al., 2025; Liu et al., 2025). The merge space produced by BPE is thus bounded by this pre-tokenization constraint. While whitespace provides reliable word boundaries in alphabetic scripts (Manning & Schutze, 1999), Chinese lacks such delimiters. Traditional Chinese NLP pipelines often rely on heuristic rules to segment text, compensating for the absence of explicit word boundaries in written Chinese. These heuristics commonly involve dictionary-based matching, statistical modeling, or rule-based systems to determine segmentation points (Wu & Tseng, 1993; Ma et al., 2005). While such approaches have achieved reasonable success, they struggle with ambiguities and out-of-vocabulary words, often resulting in inconsistent or fragmented tokenization. Dictionary-based methods are particularly sensitive to lexicon coverage, failing on novel terms, while statistical methods require extensively annotated corpora and may lack domain adaptability.

### 2.3. Information-Theoretic Cues

Information theory offers language-agnostic cues for identifying word boundaries. Early work suggested that statistical irregularities, such as peaks in mutual information and entropy, correlate with morphological structure (Light, 1996). This insight was later formalized through models based on branching entropy, which measures the uncertainty of character sequences in left and right contexts to identify likely segmentation points (Tanaka-Ishii, 2005). Such entropy-based methods have enabled effective unsupervised word segmentation, particularly in languages like Chinese (Jin & Tanaka-Ishii, 2006). To the best of our knowledge, however, these techniques have not been widely adopted in modern tokenizers for large language models.

### 2.4. Morphological and Sub-character Tokenization

Beyond word-level tokenization, recent work has advanced morphological and sub-character tokenization techniques to better capture linguistic structure and improve generalization across typologically diverse languages. For example, MorphPiece (Jabbar, 2023) is a morphologically informed tokenizer that applies an analyzer to split off any known prefixes, suffixes, and stems.

In languages with rich character composition, breaking characters into smaller linguistic units has proven effective. For Chinese, one approach (Si et al., 2023) encodes each character as a short sequence of glyph-based or phonetic components before applying subword segmentation. This results in shorter token sequences, shared representations for homophones, and strong downstream performance. Similarly, in Korean, decomposing each Hangul syllable into its constituent Jamo letters before applying byte-pair encoding yields a significantly smaller vocabulary and shorter sequences (Lee et al., 2025). Such methods retain sub-syllabic morphological information and outperform syllable-level tokenization in low-resource machine translation tasks.

### 2.5. Byte-Level Tokenization

An emerging line of research in language modeling seeks to eliminate reliance on fixed subword vocabularies by operating directly on raw byte sequences. Byte Latent Transformer (BLT) (Pagnoni et al., 2024) exemplifies this approach by dynamically segmenting input into variable-length byte patches, with boundaries guided by next-byte entropy from a lightweight language model. This enables adaptive computation and has shown performance comparable to BPE-based models at the 8B scale.

Other notable byte-level models include ByT5 (Xue et al., 2022), which processes UTF-8 byte sequences directly, removing the need for explicit tokenization. It shows strong multilingual performance and robustness to noise, particularly in low-resource settings and languages under-served by subword vocabularies. Similarly, CANINE (Clark et al., 2022) operates at the character level and introduces a downsampling mechanism to manage sequence length. Both mod-

els perform competitively with subword-based approaches, underscoring the potential of tokenization-free pipelines for language-agnostic applications.

# 3. Methods

Motivated by information-theoretic signals and the structural challenges of unsegmented scripts like Chinese, we investigate two pre-tokenization strategies for Chinese BPE based on entropy signals. Both aim to identify linguistically plausible token boundaries prior to subword vocabulary construction. The first method uses symbolic statistical measures (detailed in Algorithm 1), while the second leverages uncertainty estimates from an auto-regressive language model. In both cases, the resulting pre-segmented corpus is passed to a standard BPE tokenizer with whitespace-based pre-tokenization, thereby constraining merges to occur only within identified spans. This preserves the efficiency and scalability of BPE while biasing token construction toward linguistically meaningful units.

## 3.1. Statistical-based Pre-tokenization

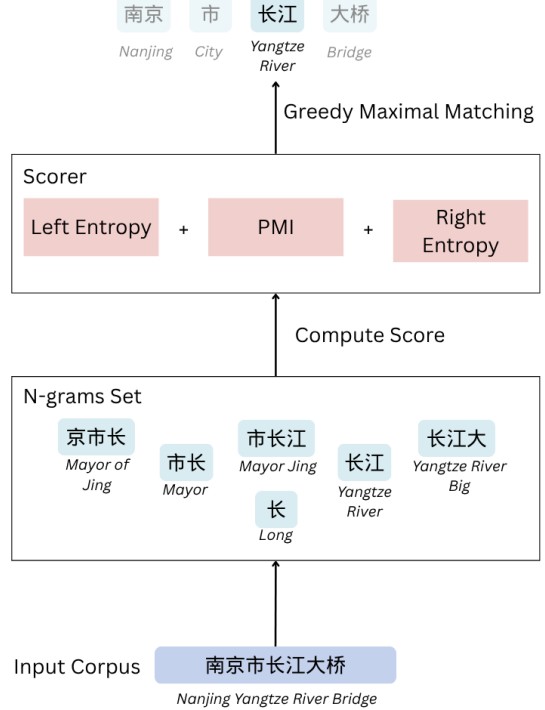

Figure 1. Overview of the statistical method. The algorithm applies greedy maximal matching based on scores from Left Entropy, PMI, and Right Entropy to select meaningful n-grams, producing the final segmentation.

The statistical method draws inspiration from unsupervised word-segmentation literature (Jiang et al., 2022). We enu-

merate every possible $n$-gram ($1 \leq n \leq n_{\max} = 6$) in the corpus and then assign a utility score to each occurring $n$-gram. Specifically, every $n$-gram is treated as a candidate $w$, and its utility score is based on a combination of internal cohesion and contextual separability:

$$U_{\text{stat}}(w) = \min_{(c_i, c_{i+1}) \subset w} \text{PMI}(c_i, c_{i+1})$$
$$+ \lambda \, \min\big(H_{\text{left}}(w), H_{\text{right}}(w)\big),$$

where $c_i$ denote the characters within $w$, and $(c_i, c_{i+1})$ denote consecutive characters. Here, we use pointwise mutual information (PMI) which measures the associative strength between two adjacent characters:

$$\text{PMI}(c_i, c_{i+1}) = \log \frac{f(c_i c_{i+1}) \, T}{f(c_i) \, f(c_{i+1})},$$

where $f(\cdot)$ denotes corpus frequency and $T$ is the total number of n-gram tokens observed in the corpus. A large PMI indicates that the pair co-occurs far more often than chance, suggesting that they should remain in the same token. We take the minimum PMI among all adjacent pairs inside $w$ so that a single weak link can lower the overall cohesion score, preventing loosely connected parts from being merged.

**Contextual separability:** Left and right entropy quantifies how diverse a span $w$ appears within its immediate context:

$$H_{\text{left}}(w) = -\sum_l P(l \mid w) \log P(l \mid w),$$
$$H_{\text{right}}(w) = -\sum_r P(r \mid w) \log P(r \mid w),$$

where $P(l \mid w) = \frac{f(lw)}{\sum_{l'} f(l'w)}$ and $P(r \mid w) = \frac{f(wr)}{\sum_{r'} f(wr')}$, with $l$ and $r$ indexing the set of distinct characters that can occur immediately to the left resp. right of $w$ in the corpus (and similar for $l'$ and $r'$). A large entropy means the span occurs with many different neighbors, signaling a plausible word boundary. We again take the minimum of left and right entropies so that a single side with low diversity keeps $w$ from being split prematurely.

**Balancing the two terms:** As illustrated in Figure 2, the ranges of PMI and entropy differ significantly, often by several orders of magnitude. This disparity can cause one score to dominate the utility score if left unadjusted. To address this imbalance, we introduce a scaling hyperparameter, $\lambda \geq 0$, which serves to modulate the relative contributions of both terms. By tuning $\lambda$, we can control the extent to which each term influences the overall optimization process, ensuring neither overwhelms the other.

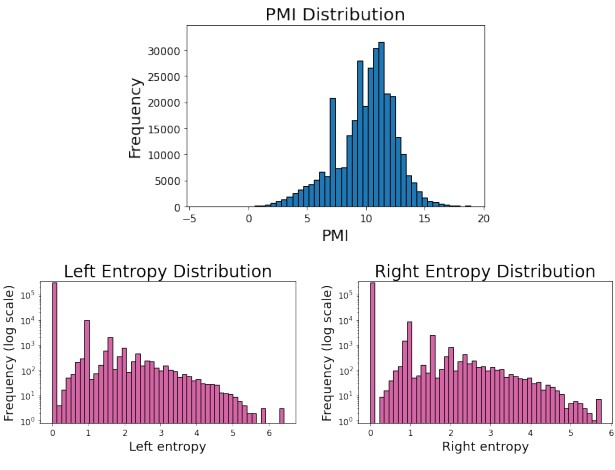

*Figure 2.* Distributions of statistical features from the PKU dataset subset. Top: PMI distribution showing a peak in the range of 9–12. Bottom: Left and right entropy distributions, shown on a log scale, are heavily skewed toward zero, indicating that many characters consistently occur within fixed contexts.

**Algorithm 1** Statistical-based Pre-tokenization

1: **Input:** Corpus $C$, maximum span length $n_{\max}$, weighting hyperparameter $\lambda$
2: **Output:** Pre-segmented corpus $C'$
3: $W \leftarrow$ all unique $n$-grams in $C$ for $1 \leq n \leq n_{\max}$
4: **for all** $n$-grams $w \in W$ **do**
5: $\quad U_{\text{stat}}(w) \leftarrow \min_{(c_i, c_{i+1}) \subset w} \text{PMI}(c_i, c_{i+1}) + \lambda \cdot \min(H_{\text{left}}(w), H_{\text{right}}(w))$
6: **end for**
7: **for all** sentence $s$ in $C$ **do**
8: $\quad C'_s \leftarrow [], i \leftarrow 0$
9: $\quad$**while** $i < \text{len}(s)$ **do**
10: $\quad\quad S_i \leftarrow \{w \in W : w \text{ starts at } i \text{ in } s\}$
11: $\quad\quad w^* \leftarrow \arg\max_{w \in S_i} U_{\text{stat}}(w)$
12: $\quad\quad$Append $w^*$ followed by a space to $C'_s$
13: $\quad\quad i \leftarrow i + |w^*|$
14: $\quad$**end while**
15: **end for**
16: Concatenate all $C'_s$ into final pre-segmented corpus $C'$

**Greedy maximal matching:** After assigning utility scores to candidate spans, the sentence is processed in a left-to-right traversal. At each character position, the span with the highest score that begins at that location is selected. Once a span is chosen, all characters it encompasses are marked as fixed, thereby preventing subsequent spans from overlapping with it. This single-pass procedure produces a non-overlapping segmentation that optimizes local utility without requiring backtracking. Characters not included in any multi-character span are treated as singleton segments. We then insert a space after every selected segment, producing a whitespace-delimited corpus. The space-delimited corpus is finally fed to a standard BPE tokenizer, which performs merge operations only within the boundaries defined by these spaces.

### 3.2. Auto-regressive LLM-based Pre-tokenization

The second approach estimates token boundaries using predictive uncertainty derived from a pretrained auto-regressive language model. At each character position $t$, we compute the conditional entropy of the next token conditioned on the tokens observed so far (up to position $t$) in the input sequence:

$$H(x_t \mid x_{<t}) = -\sum_{x \in \mathcal{V}} P(x_t = x \mid x_{<t}) \log P(x_t = x \mid x_{<t}) \tag{1}$$

The conditional entropy measures the model's uncertainty about the next character. When the value is low, the upcoming symbol is highly predictable from its left context;

conversely, sharp spikes in entropy indicate a sudden drop in predictability, signaling a likely semantic shift and the onset of a new token. Figure 3 shows the resulting entropy boundaries on sample sentences. For more samples, see Appendix A.

For this method, we use a GPT-2 model trained specifically for Chinese language modeling (Radford et al., 2019b; Zhao et al., 2019), which is open source and easily accessible on HuggingFace. Although we explored larger and more recent architectures, we found that this GPT-2 model offers a good balance between model capacity and computational efficiency. The model consists of 24 transformer decoder layers with a hidden size of 1024, yielding approximately 325 million parameters in total. During inference, we tokenize input sentences at the character level, compute the per-token entropy based on the model's output distribution, and insert segmentation boundaries at entropy peaks.

## 4. Experiments

### 4.1. Datasets

This study uses a subset of the PKU dataset from the SIGHAN 2005 bake-off task (Emerson, 2005). The PKU corpus is a widely used benchmark in Chinese word segmentation, consisting of sentences from news articles. The sentences are manually annotated by human annotators following internal guidelines for defining word boundaries in formal news text. These gold-standard segmentation boundaries provide a reliable reference for evaluating the alignment of predicted segmentation boundaries with linguistically validated ground truth.

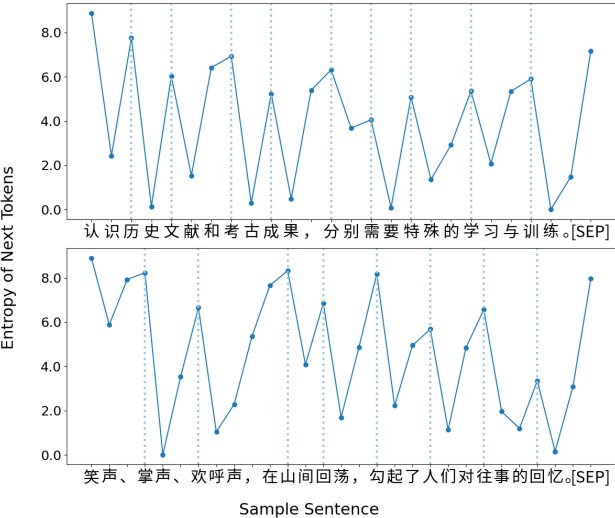

*Figure 3.* Next-character entropy scores for two randomly selected Chinese sentences evaluated by GPT-2. Each plot illustrates the entropy of the model's next-character prediction at each token position. Blue dashed lines denote local peaks, which serve as span boundaries. These examples are provided to illustrate how the model's uncertainty varies across different parts of a sentence.

Due to the computational cost associated with pre-tokenization entropy calculation using the autoregressive method, which required approximately two days on two NVIDIA A100 GPUs, we limited our experiments to 10% of the full PKU training corpus. This subset consists of 2,255 sentences and approximately 90,000 Chinese characters. All characters are in simplified Chinese, and the original corpus is split into sentences based on ending punctuation.

### 4.2. Baseline and Configurations

We evaluate three tokenization strategies, each based on Byte-Pair Encoding (BPE). Prior work suggests that a smaller vocabulary size is more suitable for Chinese character-based models (Li et al., 2019). For example, the GPT-2 model used in Section 3.2 has a vocabulary size of 21,128 tokens. Given the size of our dataset, we chose a reduced vocabulary of 12,000 tokens to balance representational efficiency and data sparsity. All methods operate on the same input sequences derived from the preprocessed PKU corpus described in Section 4.1. Our three methods are as follows:

- **Standard BPE**: A baseline implementation of frequency-based BPE applied directly to character sequences, without any pre-tokenization. This mirrors the standard application of BPE in languages like Chinese, where whitespace-based tokenization is not applicable. Merges are selected purely based on adja-

cent symbol pair frequency, without any linguistic constraints.

- **Statistically-based Entropy + BPE**: Our method from Section 3.1 that introduces a pre-tokenization step based on statistical informational cues. Character sequences are first segmented using a score that combines PMI and left/right entropy. The resulting boundaries constrain BPE merges to occur only within identified spans.

- **Auto-regressive LLM-based Entropy + BPE**: Our method from Section 3.2 that applies pre-tokenization using next-character predictive entropy estimated from a pretrained autoregressive language model. Local entropy maxima are used as indicators of segmentation boundaries. These boundaries are inserted into the sequence prior to BPE training.

### 4.3. Qualitative Segmentation Analysis

To better understand the behavior of entropy-guided pre-tokenization, we present a qualitative visualization of token boundary selection under different methods. This analysis provides insight into how statistical and model-based entropy signals influence segmentation decisions prior to BPE application.

Figure 4 illustrates the token boundaries selected by multiple pre-tokenization strategies for a representative Chinese sentence. Each row corresponds to a different method: the gold-standard segmentation from the PKU dataset, segmentation based solely on predictive entropy from GPT-2, segmentation using left/right entropy alone (entropy-only), and the statistical method with varying values of the weighting parameter $\lambda$ (from top to bottom). We explored $\lambda$ values using a standard grid search. Vertical lines denote the identified segmentation boundaries.

This visualization highlights several key trends. First, the predictive entropy method identifies boundaries that align closely with semantic units, often matching human annotations. Second, the statistical method demonstrates flexible boundary control via the $\lambda$ parameter; smaller values result in shorter, more fragmented tokens, while larger values emphasize contextual diversity, yielding longer and more coherent spans. Notably, the entropy-only method achieves reasonable alignment with the gold standard, suggesting that information-theoretic signals alone carry substantial linguistic relevance even without subsequent BPE processing.

These observations support the hypothesis that entropy-informed pre-tokenization can act as a lightweight yet effective proxy for unsupervised word segmentation, offering a principled mechanism to introduce structure into the BPE pipeline for unsegmented scripts.

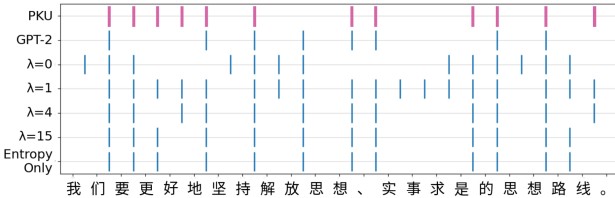

*Figure 4.* Comparison of pre-tokenization methods on a sample Chinese sentence. Observing the vertical lines from top to bottom, the method with $\lambda = 4$ yields token boundaries that align most closely with the gold standard in the top row. It segments nearly perfectly compared to the ground truth in the top row, with only one extra boundary inserted after the 10th character.

## 4.4. Intrinsic Evaluation

We assess segmentation quality using precision, recall, and F1 score by comparing predicted subword boundaries to gold-standard word segmentation in the PKU corpus. Precision reflects the proportion of predicted boundaries that align with true word boundaries, while recall measures the proportion of true boundaries that are correctly predicted. The F1 score is computed following the official SIGHAN bake-off evaluation script (Emerson, 2005), which defines a word-level match as a segment whose start and end positions exactly align with a gold-standard word. We split our corpus into 70% training and 30% testing for model development and evaluation.

Our experimental procedure consists of four main steps:

1. **Pre-tokenization:** We apply various pre-tokenization strategies to the training data. These include entropy-based methods, GPT-2 uncertainty, and a no-pre-tokenization baseline.

2. **BPE Training:** A Byte-Pair Encoding (BPE) tokenizer is trained on the pre-tokenized training set. All methods use the same algorithm and trainer configuration.

3. **Segmentation:** The trained tokenizer is then used to segment the test set at inference time. The resulting tokens are treated as predicted word boundaries.

4. **Evaluation:** We compute precision, recall, and F1 score by comparing the predicted word boundaries to the PKU gold-standard segmentation.

Our results in Table 4.4 demonstrate that entropy-based pre-tokenization methods outperform the baseline BPE approach. The best performance is achieved when using an entropy-regularized approach with $\lambda = 4$, which yields the highest F1 score of 58.73, significantly surpassing the baseline's 49.30 by 9.43 percentage points. This setting also achieves the best precision (54.21) and the second-highest

recall (64.06), indicating its effectiveness in correctly identifying subword boundaries with fewer false positives.

| Method | Precision | Recall | F1 |
|---|---|---|---|
| Baseline | 46.89 | 51.96 | 49.30 |
| GPT-2 | 52.07 | **64.69** | 57.70 |
| $\lambda = 0$ | 28.69 | 42.92 | 34.39 |
| $\lambda = 1$ | 41.24 | 55.91 | 47.47 |
| $\lambda = 4$ | **54.21** | 64.06 | **58.73** |
| $\lambda = 15$ | 52.83 | 62.17 | 57.12 |
| Entropy Only | 51.28 | 60.98 | 55.71 |

*Table 1.* Segmentation results on the PKU dataset. All scores are computed on the test split containing 30% of 2,255 sentences using character-level boundary comparison.

The GPT-2 method, which leverages language model uncertainty for boundary detection, performs competitively, achieving an F1 score of 57.70 and the highest recall (64.69), though its precision (52.07) is slightly below that of $\lambda = 4$. These results highlight the strong predictive signal of token-level entropy derived from pretrained LLMs, even without additional regularization.

The entropy-only method also shows strong performance (F1 = 55.71), suggesting that raw entropy is a useful heuristic for segmentation. However, it is outperformed by the regularized variants, indicating that combining entropy with structural constraints improves segmentation accuracy.

Varying the regularization strength $\lambda$ provides insight into the trade-off between precision and recall. A low value like $\lambda = 0$ yields poor overall performance (F1 = 34.39), primarily due to low precision (28.69). In contrast, moderate values such as $\lambda = 1$ and $\lambda = 15$ show solid gains (F1 = 47.47 and 57.12, respectively), with $\lambda = 15$ emphasizing recall (62.17) more than precision.

Overall, these results show that entropy-based pre-tokenization with tuned parameters provides the most accurate and balanced subword boundary predictions, outperforming both frequency-based baselines and entropy boundaries derived from auto-regressive language models.

## 5. Conclusion

This paper introduces two entropy-driven pre-tokenization methods to address the limitations of Byte-Pair Encoding in unsegmented languages such as Chinese. By incorporating information-theoretic signals, specifically statistical co-occurrence metrics and predictive entropy from a pretrained autoregressive language model, we effectively bias BPE toward more linguistically coherent token boundaries. Experimental results on the PKU segmentation benchmark confirm that both approaches significantly outperform standard frequency-based BPE, with the statistical method ca-

pable of yielding the highest F1 score with proper hyperparameter tuning. Our methods preserve the modularity and efficiency of existing BPE frameworks while improving token granularity and interpretability.

While this work focuses on tokenizer accuracy within an annotated dataset, an important direction for future research is to integrate these pre-tokenization methods into large language model (LLM) training and evaluate their impact on downstream tasks such as machine translation and named entity recognition. Our findings suggest that these strategies could be particularly beneficial for modeling low-resource or unsegmented languages within standard transformer-based frameworks. Another promising avenue is adapting these methods for byte-level tokenization, which has gained popularity in multilingual settings for its robustness and language independence. Embedding structural cues into byte-level token streams may enable models to retain the generality of byte-based representations while incorporating morphological awareness.

## Acknowledgments

We would like to thank Alihan Hüyük and Weiwei Pan for helpful guidance with this project, and the reviewers for their time and effort.

## Impact Statement

This paper presents work aimed at advancing the field of Natural Language Processing (NLP) by improving models' ability to understand unsegmented text, thereby contributing to more equitable language technologies that extend beyond English. Given that all tokenization decisions directly influence model behavior and downstream task performance, our proposed methods should be critically evaluated in applied contexts before deployment. While our work seeks to enhance the performance of NLP systems across diverse languages, we are not aware of any immediate societal concerns that necessitate specific discussion.

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

# A. Additional Entropy Visualizations of GPT-2 Outputs

To further illustrate character-level entropy, Figure 5 presents entropy distributions for additional sample sentences from the PKU dataset. Each subplot corresponds to a single sentence, with segmentation boundaries, indicated by dotted vertical lines, aligned to local entropy maxima computed with the GPT-2 model. For instance, the segmentation of the first sentence yields the token sequence: ['岁月的', '流逝', '显得', '凄美而', '沉毅，', '荣辱', '与成败', '交替，', '诞生与', '消亡', '轮回。']. Note that entropy plots of this kind are not applicable to the statistically-based pre-tokenization method (Section 3.1), as its segmentation relies on iterative $n$-gram merging rather than character-level entropy.

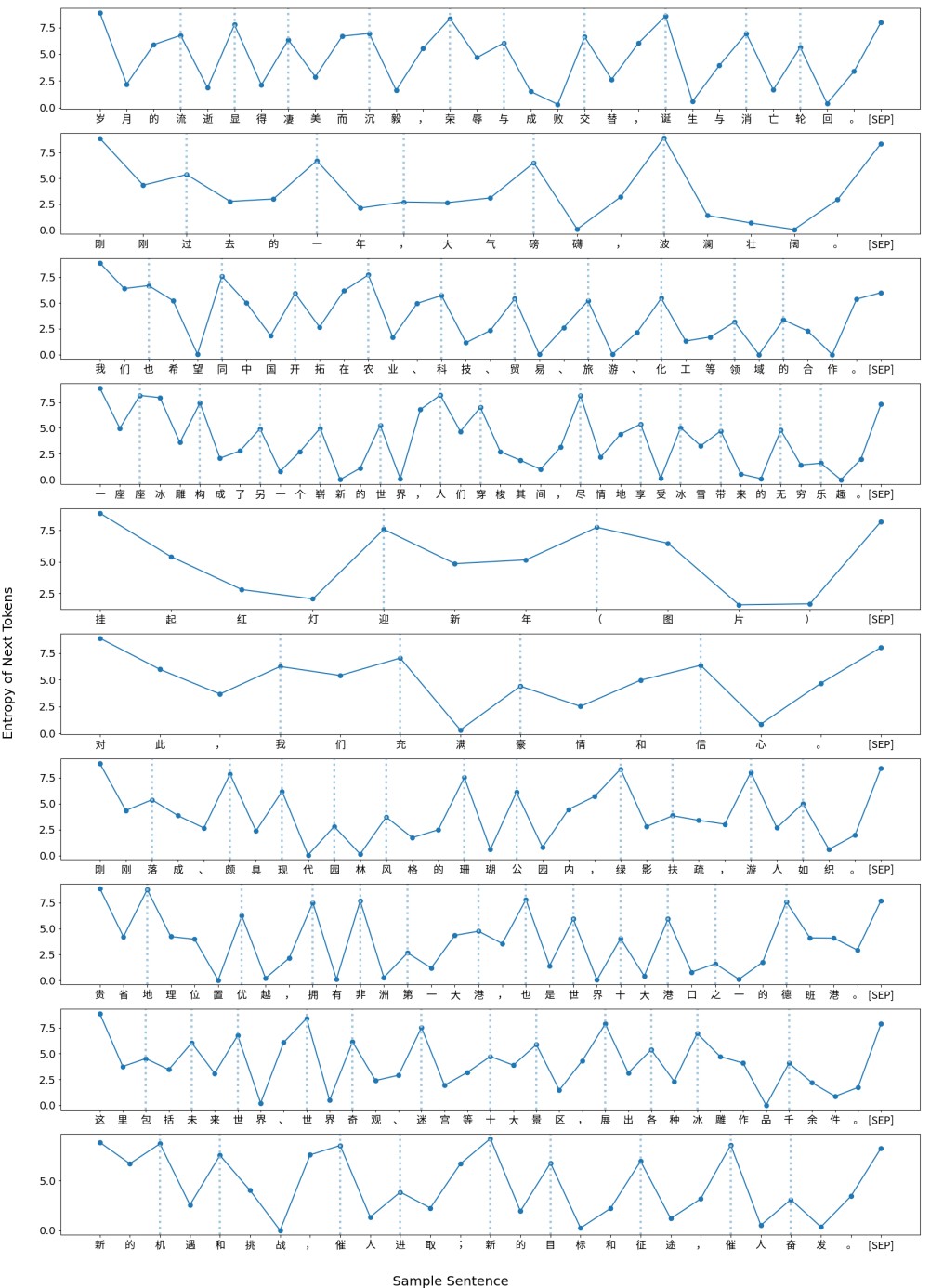

*Figure 5.* Next-character entropy scores for additional sample sentences from the PKU dataset.

