# OpenReview forum: "Entropy-Driven Pre-Tokenization for Byte-Pair Encoding"
_ICML.cc/2025/Workshop/TokShop — TokShop_

### Official Review · Reviewer_3nB2 · 2025-06-06
**The paper proposes a method to improve BPE pre-tokenization with entropy.**

**Rating:** 7
**Confidence:** 4

**Review:**

The paper proposes a method to improve BPE pre-tokenisation with entropy. This is done in two ways:
* by scoring all n-grams in the corpus with a utility score that combines entropy and the minimal mutual information between characters in the n-gram
* by computing the entropy of the next token with a GPT-2 model trained for Chinese language modelling.

The authors test the method on the  PKU dataset and see that it produces more linguistically meaningful tokens.

**Strengths**: The related works are well explained. The main ideas of the methods are clear.

**Weaknesses**: Some terms in the formulas are not well explained  (e.g., “r” and “l” in Eq.3-4. Are they the right/left side of the n-gram or of the entire sentence?)

**Minor issues**:
* Typo line 157: ljeft
* Figure labels are too small and difficult to read
* It would be interesting to see more output examples

This paper [1] (and this other paper [2] cited by [1]) might be relevant, since they also investigate how to use entropy for tokenisation.

[1]: [Efficient Transformers with Dynamic Token Pooling](https://aclanthology.org/2023.acl-long.353/) (Nawrot et al., ACL 2023)
[2]: [Finding Structure via Compression](https://aclanthology.org/W98-1210/) (Hutchens & Alder, CoNLL 1998)

---

### Official Review · Reviewer_nMeU · 2025-06-06
**Pretokenization for Chinese**

**Rating:** 4
**Confidence:** 4

**Review:**

This paper presents a novel pre-tokenization strategy, particularly targeted to (simplified) Chinese. In fact, the authors present two such strategies: one based on the entropy of the left/right context of bigrams, the other based on the perplexity of a language model (GPT2 here).
After this pre-tokenization, standard BPE is applied. The different methods are evaluated by comparing final tokens to gold tokens (a question below on what this means).

Tackling pre-tokenization strategies is hugely important, as current default decisions are inspired by latin alphabets which are easily pre-tokenized using whitespaces.

My biggest concern is on the use of a single intrinsic metric to measure how good a tokenization stragey is. In particular, I do not agree with the motivation that `created token sequences may not align with linguistically meaningful units, which can degrade downstream performance and interpretability.` The statement that tokens have to be linguistically meaningful units is at worst wrong, at best contentious. While there are papers on both sides of that debate, the authors should support this claim with some strong results.

Another concern is that the approached is tested only on one language. Considering that the original motivation comes from the fact that existing approaches are not optimized for Chinese, it seems only fair to ask for a comparison on some whitespace-based language.

Other questions:

- What if during inference a new token appears?

- `the gold-standard segmentation from the PKU dataset`: is this based on a linguistic parser? If so, which?

- why not just use this gold standard (assuming it can be automatically computed) to pre-tokenize the text?

- do you have other references on the use of entropy of the left/right context? I am only aware of this rather theoretical work ([xkcd-repeats](https://www.sciencedirect.com/science/article/pii/S1570866717300655))

---

### Official Review · Reviewer_pAJS · 2025-06-07
**The paper studies entropy-based pre-tokenization for Chinese; the approach is evaluated on a benchmark for segmentation and performs better than BPE.**

**Rating:** 7
**Confidence:** 3

**Review:**

The paper proposes two entropy-informed pre-tokenization strategies for Chinese as a basis to guide BPE segmentation. Chinese, as an unsegmented language, is challenging for BPE as it provides no cues on linguistic boundaries.

The first strategy relies on pointwise mutual information and left/right entropy to identify likely segmentation points.
The second strategy is derived from predictive entropy from a GPT2 model to detect boundary uncertainty.

The proposed strategies rely on the intuition that that peaks in mutual information and entropy correlate with morphological structure, and thus can provide cues for segmentation.

The experiments are carried out on a part of the PKU dataset, a benchmark for segmentation.
The authors compare standard BPE with statistically-based entropy + BPE as well as LLM-based entropy + BPE.

Strengths:
The paper presents an interesting strategy to improve the quality of Chinese segmentation; the experiments show improvement over standard BPE.

Weaknesses:
- the authors considered only a small dataset (2255 sentences) for their experiments.
- while the authors showed an improvement in an intrinsic evaluation, it is unclear whether there this also applies to a downstream task like translation.
- Similarly, it might be interesting to see the performance of the proposed strategy for a different language

Comments:
Figure 4: can you also show the BPE segmentation, as well as glosses?

Typo: 'ljeft' at the bottom of page 3

The paper "Incorporating Context into Subword Vocabularies" (Yehezkel and Pinter, EACL 2023)
might be interesting for related work.

Question:
for the LLM-based entropy, do you think the underlying segmentation of the pretrained model has an impact on the entropy?

---

### Decision · Program_Chairs · 2025-06-10

Accept